# The Use of the Kahoot! Learning Platform as a Type of Formative Assessment in the Context of Pre-University Education during the COVID-19 Pandemic Period

**Florentina Toma** [1] , **Daniel Constantin Diaconu** [2,3,]* and **Cristina Maria Popescu** [4]

[1] Simion Mehedinți "Nature and Sustainable Development" Doctoral School, University of Bucharest, 061071 Bucharest, Romania; florentina.ghersin@drd.unibuc.ro
[2] Department of Meteorology and Hydrology, Faculty of Geography, University of Bucharest, 061071 Bucharest, Romania
[3] Research Center for Integrated Analysis and Territorial Management, University of Bucharest, 4-12 Regina Elisabeta Avenue, 030018 Bucharest, Romania
[4] Viilor Economic College, 050157 Bucharest, Romania; cristinap30@yahoo.com
* Correspondence: daniel.diaconu@unibuc.ro

**Abstract:** The present study aims to display how using a personal assessment environment based on the interactive Kahoot! platform actively supports the teaching–learning process. The goal is to improve the instructive–educational process by applying a learning platform based on play and digital technology that favors a qualitative educational endeavor. The use of the Kahoot! platform as form of assessment had a significant and direct positive effect on the educational process during the COVID-19 pandemic.

**Keywords:** standardized Kahoot! assessment; game-based learning; COVID-19

## 1. Introduction

With the onset of the COVID-19 pandemic, many schools have had to approach an enforced online or hybrid teaching–learning–assessment system. Many teachers have been put in the situation of having to associate digital competences to the teaching subject in order to ensure that the educational process is being conducted in a good and effective way.

In this regard, there has recently been a large amount of research conducted on the learning, teaching, and assessment approaches that have been adopted since the outbreak of the COVID-19 pandemic and the impact of those approaches on students [1].

Several articles have demonstrated that digital learning has a beneficial influence on the intellectual development of students and teachers [2,3]. For example, the technical skills of teaching staff have increased as a result of digital learning [4]; the average duration of courses is lower [5]; the education system is better structured [6]; there has been an increase in e-learning techniques [7]; and more opportunities to give feedback to students have arisen [8].

Other studies have highlighted the inability of some schools to use online learning systems [9,10], while other research has demonstrated that some students have no competence or self-control when trained face to face [11]. A recent study also demonstrated that internet connectivity is very poor in China, with a decision being made to allow international students to remain in their student hostels in order to access their university's internet services [12].

An article produced in the context of COVID-19 used the flipped classroom (FC) teaching model, which revolutionized the field of education, did not show sufficient evidence of its advantages and disadvantages in the university setting [13].

The COVID-19 pandemic has led to changes in the development of educational processes [14]. The health rules that have been adopted at different times of the pandemic

have led to travel restrictions, physical distancing, health testing, and the vaccination of teachers, and, in certain situations, the vaccination of students over 16 years of age [15].

In many countries as well as in Romania, the transition to an online or hybrid (blended) education system has highlighted the lack of infrastructure that is needed to support these systems [16]. Central and local authorities have made efforts to acquire systems to provide the necessary logistical support (IT systems, interactive whiteboards, video projectors, audio-video systems, internet connectivity) [17].

During courses conducted in the online or hybrid format, there a lack of IT skills across teachers has been demonstrated, as many teachers could not easily translate their lessons into the digital format nor could they make choices regarding the teaching and evaluation methods that would be appropriate to this format [18].

Research exists on how a lack of motivation can lead to a reduction in learning outcomes and an uncomfortable atmosphere in the classroom [19] as well as on how experiencing boredom while on a the computer can lead to problematic behavior [20]. Educational research has shown that students who are actively involved in learning will learn more than passive students [21,22].

When considering how to increase student performance and improve teaching strategies, we find a guide for teachers in the field literature that studies visible learning [23] and a study that explores the value of computer games as a tool for student learning leading to the development of ICT skills [24].

Studies have shown us that the use of digital technology has been indicated to be suitable for measuring e-learning tools [25].

Learning technology also presents game-based learning method because it motivates and engages students in such a way as to learn without being aware of it [26], making it beneficial for the motivation and dynamics of the class [27].

Kahoot! is a free learning platform that is based on play and educational technology that can be used to create interactive tests. It can be accessed using any device, computer, or laptop that has a web browser, making the platform very accessible to students [28,29].

Kahoot!, a game-based learning platform, can be used to review student knowledge and for formative assessment. It is the most popular gaming-based learning platform, and studies have also shown how Kahoot! affects the learning performance, class dynamics, attitudes, and perceptions of students and teachers [30]. The essential requirements for Kahoot! are for teachers to create their own content and to evaluate their students, and for students to have fun, be competitive, and learn [31], thus improving the educational, learning, and assessment environment in class [32].

Most studies on Kahoot! have focused on class dynamics [33], on the effects of learning [34], on the perceptions of students and teachers regarding the game-based learning platform [35,36], and on the fact that this game increases grades and the motivation of students to learn and study the subject being taught [37].

Starting from the study about the evaluation of student results when using the interactive platform Kahoot!, the research conducted here aims to investigate whether evaluation during the feedback sequence of the lesson in the online model versus the in traditional teaching–learning–evaluation model improves student results and increases student motivation.

The aim of the current research is to study the usage of this game-based learning platform in the educational context created by the COVID-19 pandemic.

## 2. Materials and Methods

The research was conducted between April and June 2021 within the "Mihai Eminescu" National College, Bucharest, Romania, and consisted of the use of the interactive exercise Kahoot! as a form of standardized assessment for the feedback sequence of the lesson over a period of five weeks in an online teaching–learning–evaluation model and over another five weeks in a traditional model (with one week of summative assessment) on a sample of 392 students.

The groups of students used in the research are intellectually homogeneous and have a predominantly visual learning style. A Google Forms questionnaire was conducted for students regarding their perceptions of the interactive game Kahoot! for the feedback/summative assessment sequence in geography lessons.

The study was thus divided into three stages of research:

- Step 1: A study was conducted on the standardized Kahoot! interactive exercise feedback assessments for all classes in the online teaching–learning–assessment model over five weeks of school.
- Step 2: A study of the standardized feedback and final review assessments was conducted on the Kahoot! learning platform for all classes in a traditional teaching–learning–assessment model during six weeks of school.
- Step 3: Conduct, complete, and analyze the Google Forms questionnaire for student perception of the interactive game Kahoot! for the feedback/summative assessment sequence in geography lessons.

In the 1st and 2nd phases of research, all of the students in the class had provide written answers in the Kahoot! game in the last 5–10 min of the class as a feedback sequence to 9 standardized questions according to the 5 proposed operational objectives and in accordance with the 3–5 specific competencies in the school curriculum (as an assessment scale: 9 marks in total for the questions and a mark was awarded ex officio or to tests in the online model, with a feedback question on the lesson taught by the teacher). The items used on the Kahoot! platform was of a dual choice objective type (true/false) and multiple choice/quiz (where only four answers were present and where only one choice was correct).

In stage 3, student perceptions on the use of Kahoot! As an assessment tool were collected by completing a Google Forms questionnaire and was conducted mostly on a Likert scale, with the aim of building the self-correction and self-assessment capacities of the students and of determining the likelihood of the teacher using this tool as a form of assessment again in the future. This created a database of the assessment results for each class.

The methodology used to test the hypothesis of the effect scale that was studied was the quantitative statistical calculating method to determine the expected progress that the students should have made during the research period for each learning model.

## 3. Results

In the first phase of research, standardized assessments were applied on the Kahoot! learning platform for five class hours where geography was being studied (in the 5th, 6th, 9th and 10th grades) in order to observe the level of knowledge acquired in the online teaching–learning–assessment model. The results obtained by the students in each class were synthesized and are illustrated in Figure 1 (5th grade), Figure 2 (6th grade), Figure 3 (9th grades), and Figure 4 (10th grade) and reflect a similar level of knowledge for all of the classes included in the research.

In the second stage of research, standardized assessments with the Kahoot! game were also applied for six class hours where geography was being studied (in the 5th, 6th, 9th, and 10th grades) in order to observe the level of knowledge acquired in the traditional and online teaching–learning–assessment models compared to the online only model. The results obtained by the students in each class were synthesized and are reflected in Figure 5 (5th grade), Figure 6 (6th grade), Figure 7 (9th grade), and Figure 8 (10th grade), which reflect a similar level of knowledge for all of the classes included in the research.

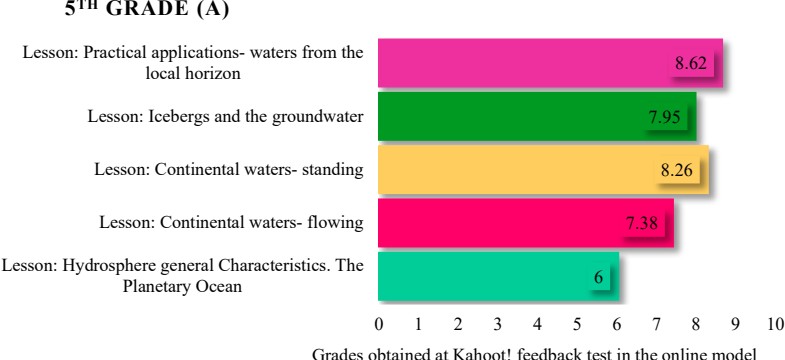

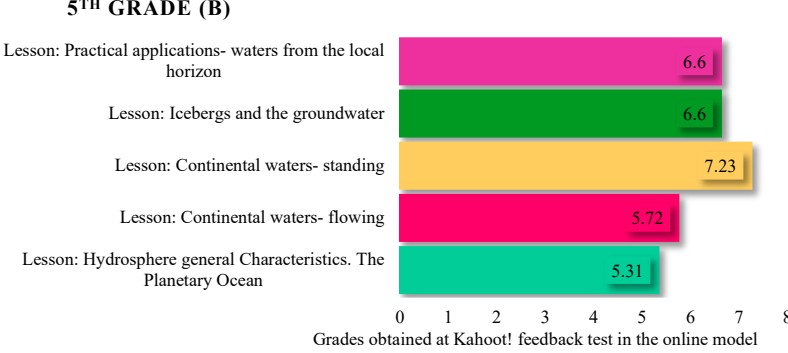

**Figure 1.** Evolution of student grades in the assessment sequence of the Kahoot! interactive game; type—traditional model of teaching–learning–assessment—5th grade parallel classrooms A and B.

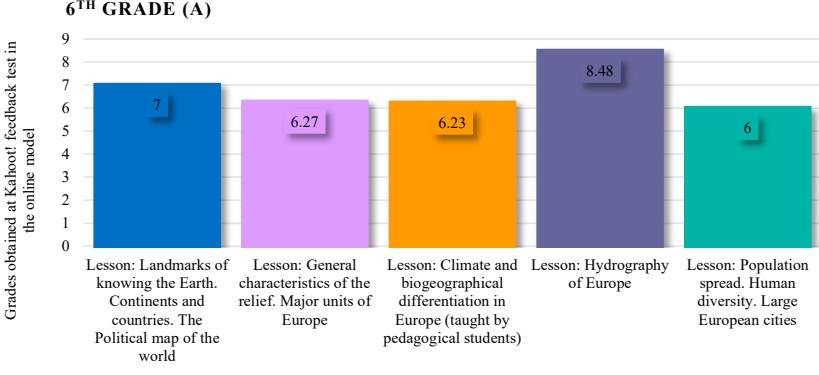

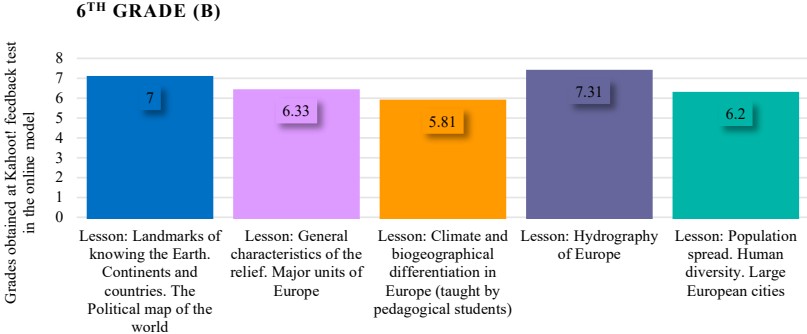

**Figure 2.** Evolution of student grades in the assessment sequence of the Kahoot! interactive game; type—traditional model of teaching–learning–assessment—6th grade parallel classrooms A and B.

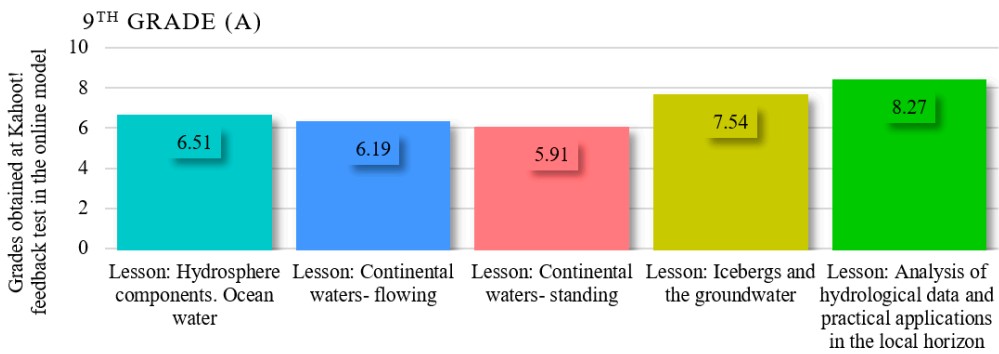

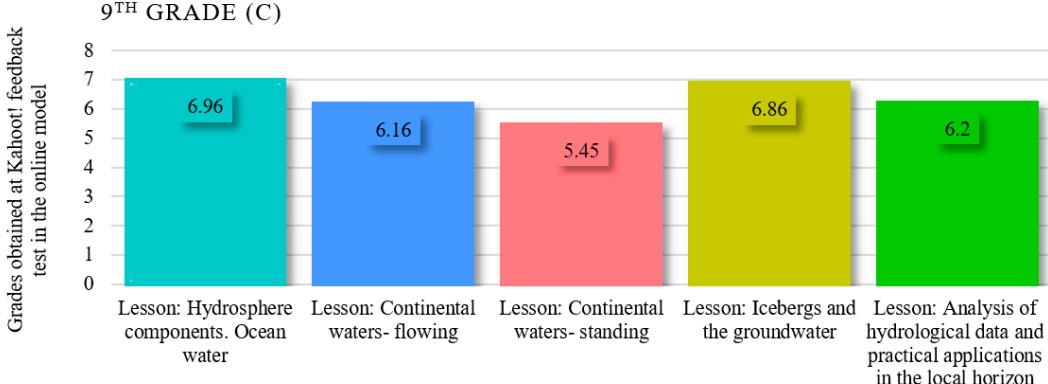

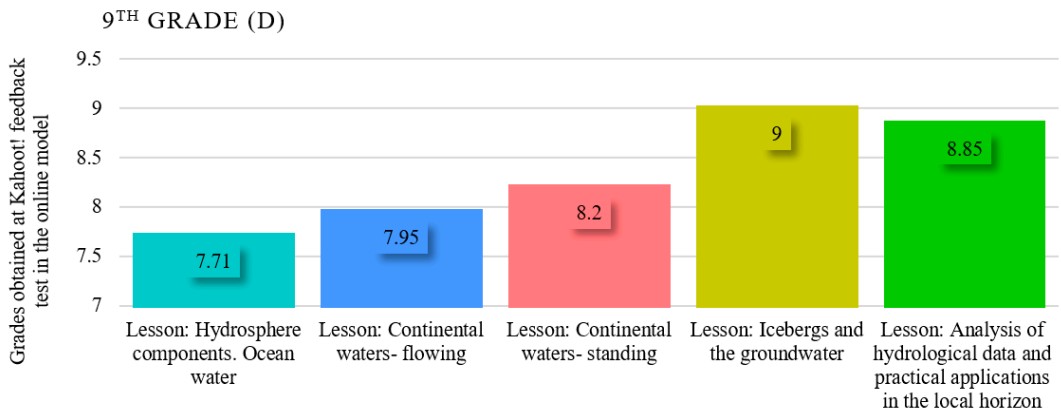

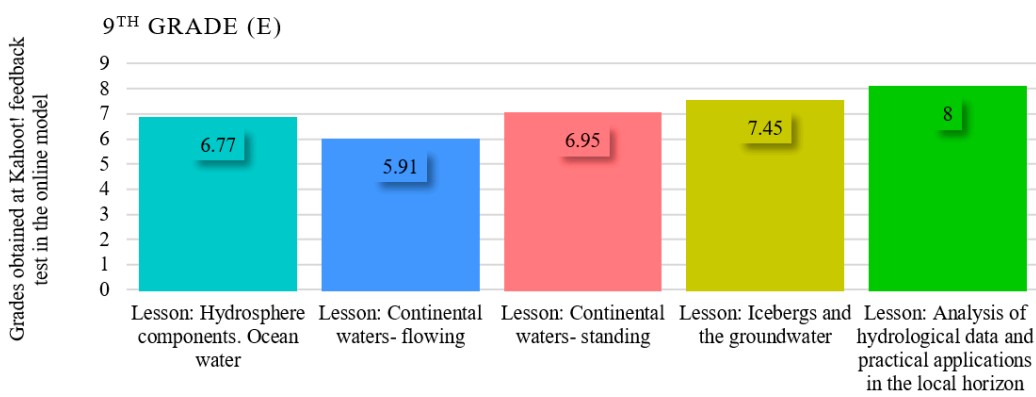

**Figure 3.** *Cont.*

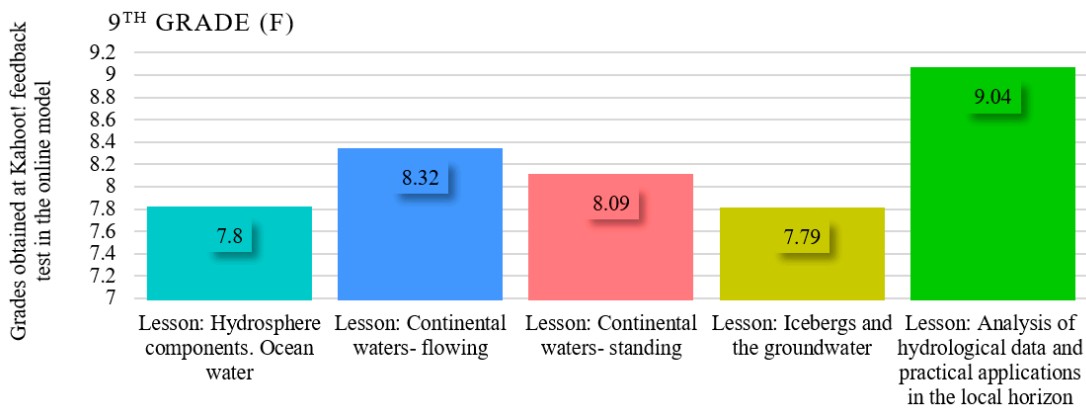

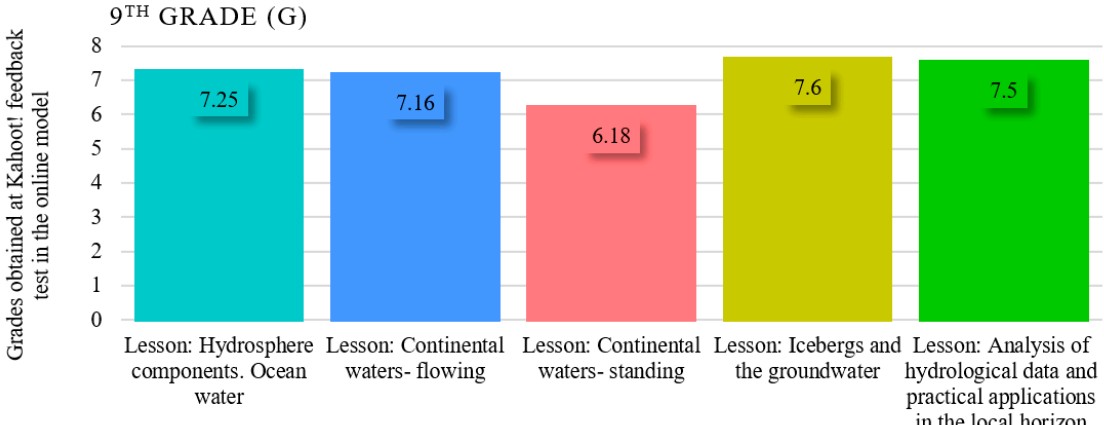

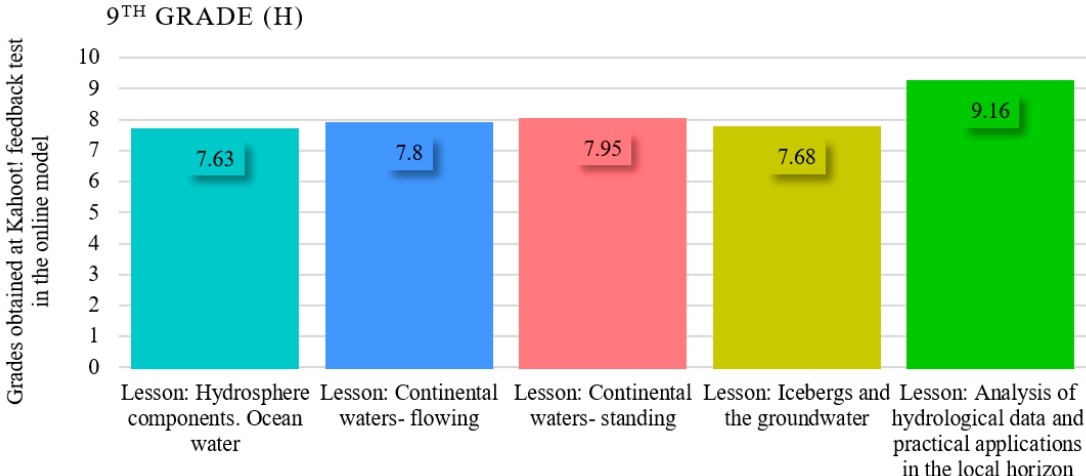

**Figure 3.** Evolution of student grades in the assessment sequence of the Kahoot! interactive game; type—traditional model of teaching–learning–assessment—9th Grade parallel classrooms A, C, D, E, F, G, H.

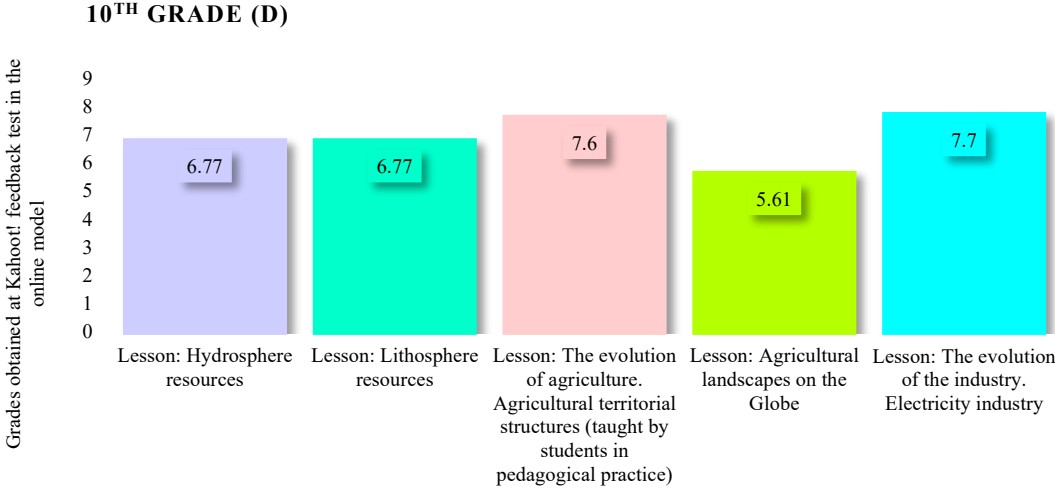

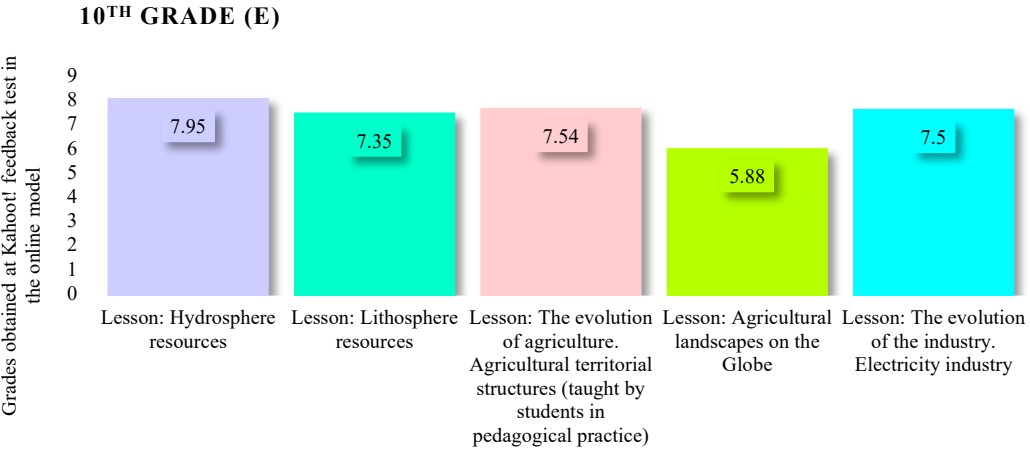

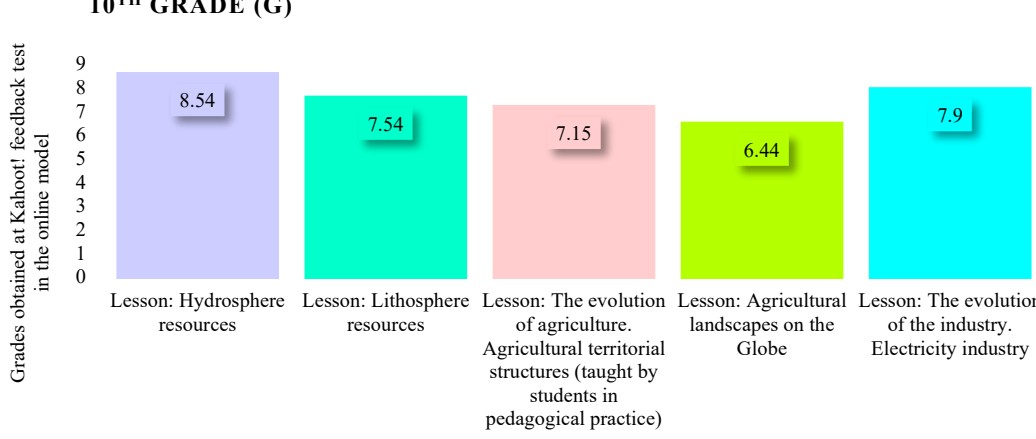

**Figure 4.** Evolution of student grades in the assessment sequence of the Kahoot! interactive game; type—traditional model of teaching–learning–assessment—10thGrade parallel classrooms D, E and G.

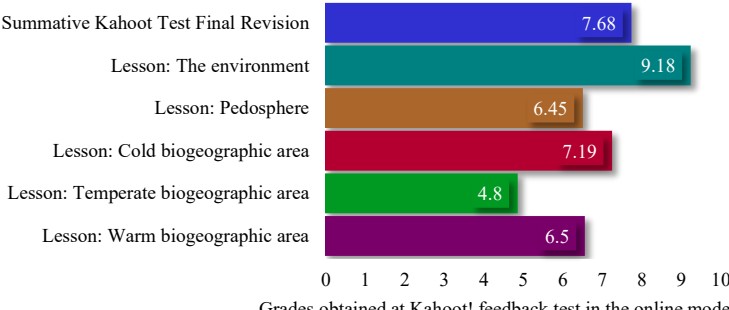

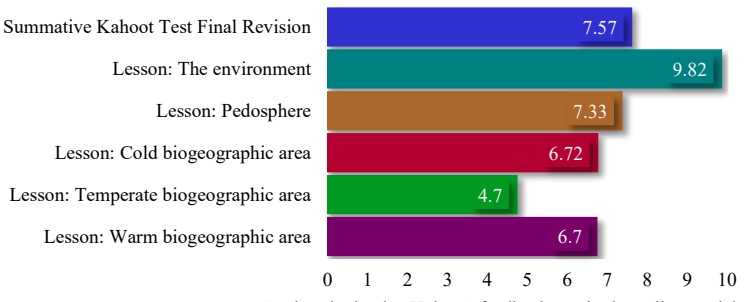

**Figure 5.** Evolution of student grades in the assessment sequence of the Kahoot! interactive game; type—traditional model of teaching–learning–assessment—5th grade parallel classrooms A and B.

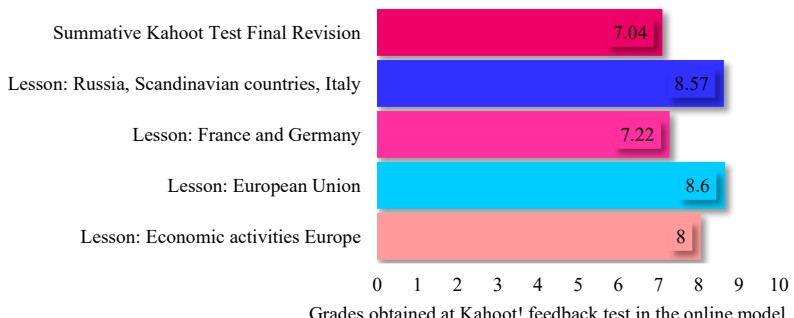

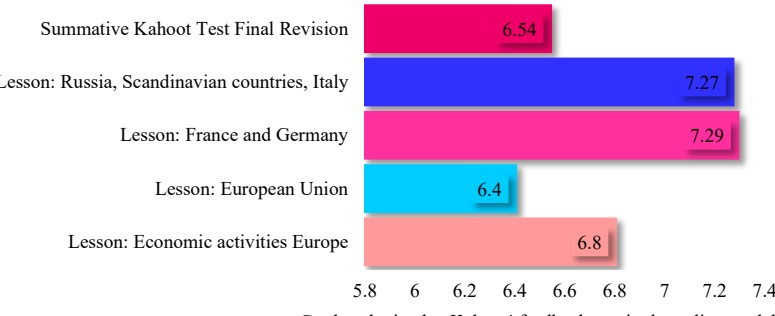

**Figure 6.** The evolution of the students to the assessment sequence of the Kahoot! interactive game; type—traditional model of teaching–learning–assessment—6th grade parallel classrooms A.

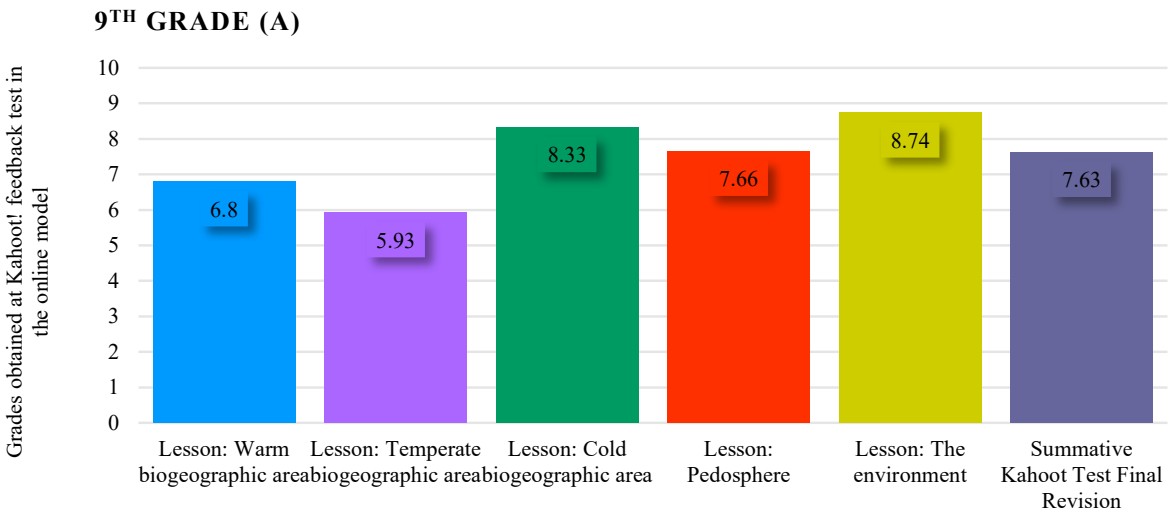

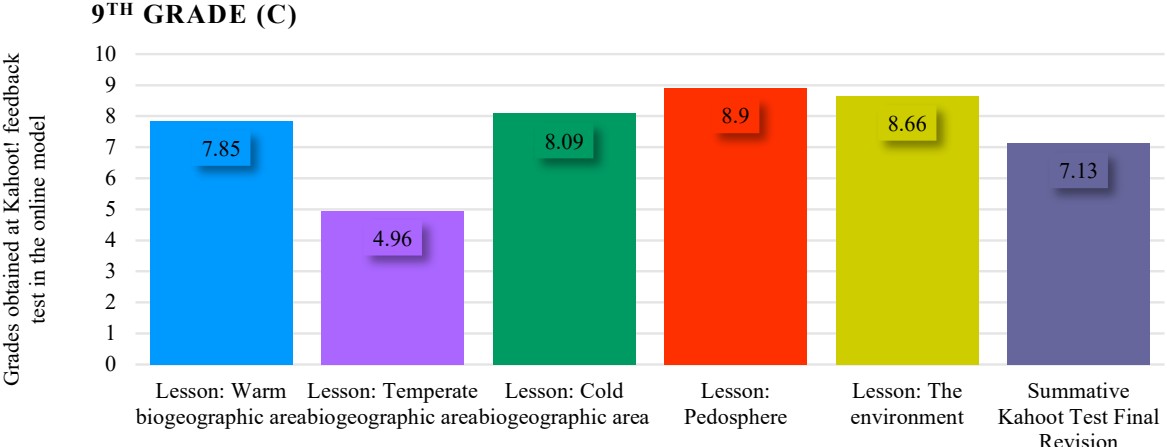

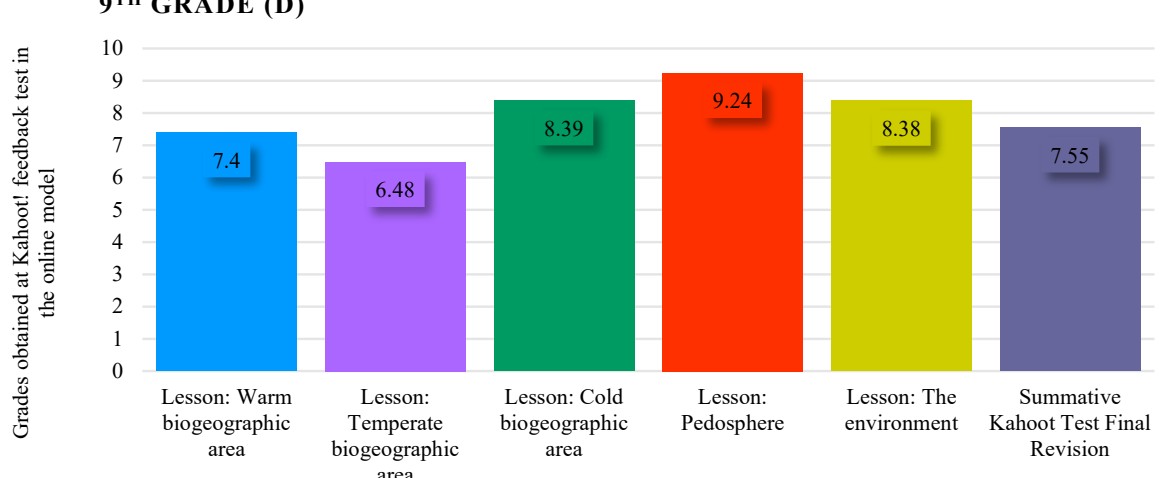

**Figure 7.** *Cont.*

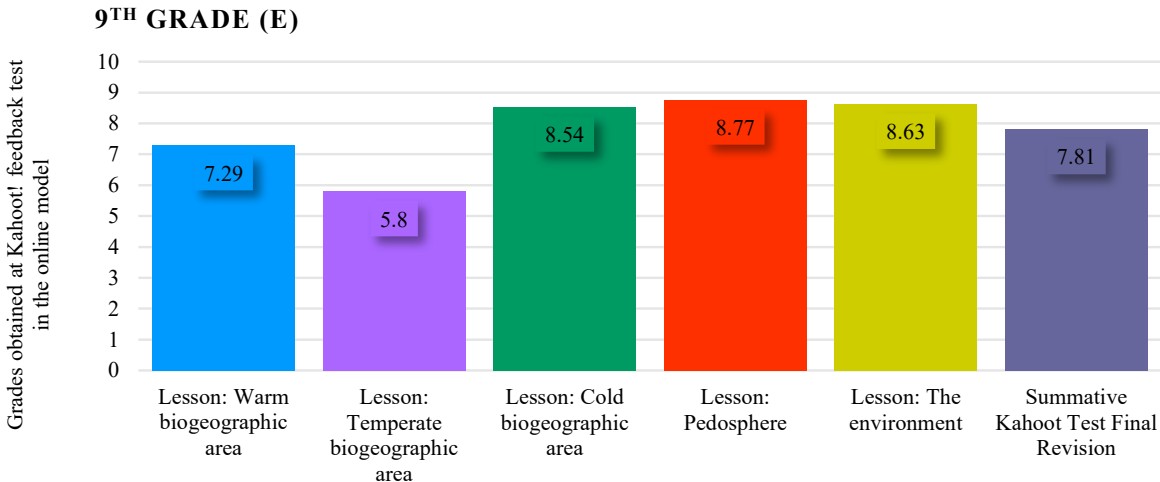

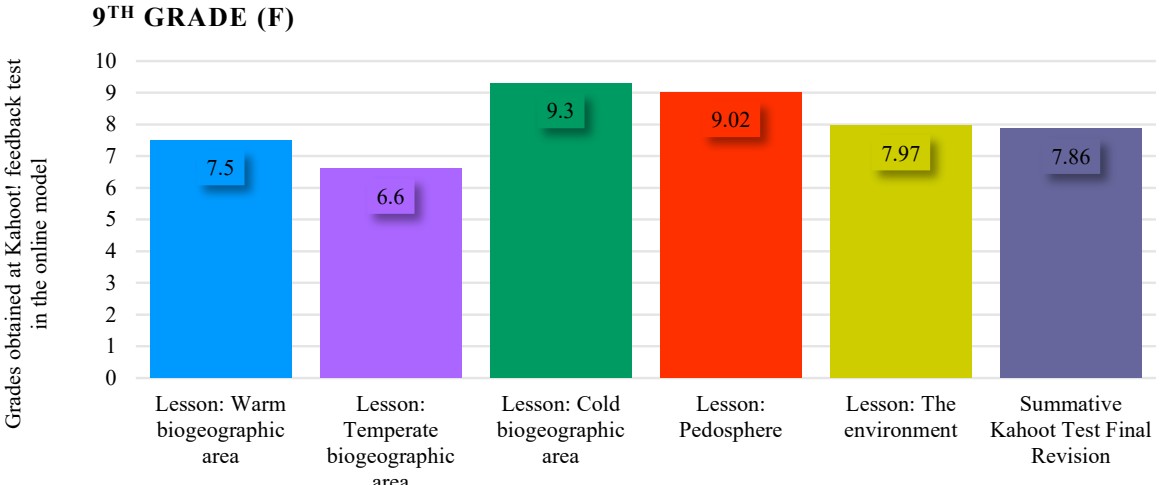

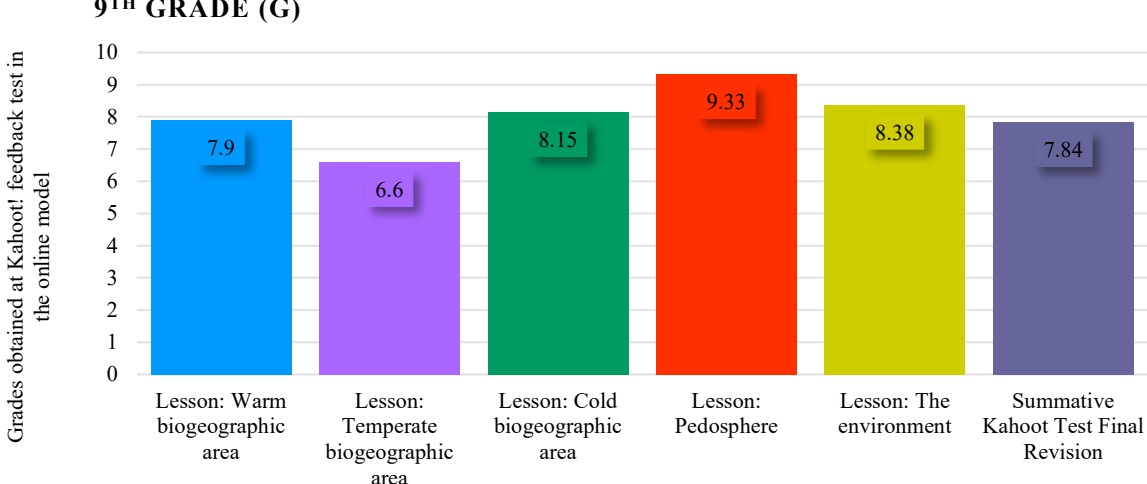

**Figure 7.** *Cont.*

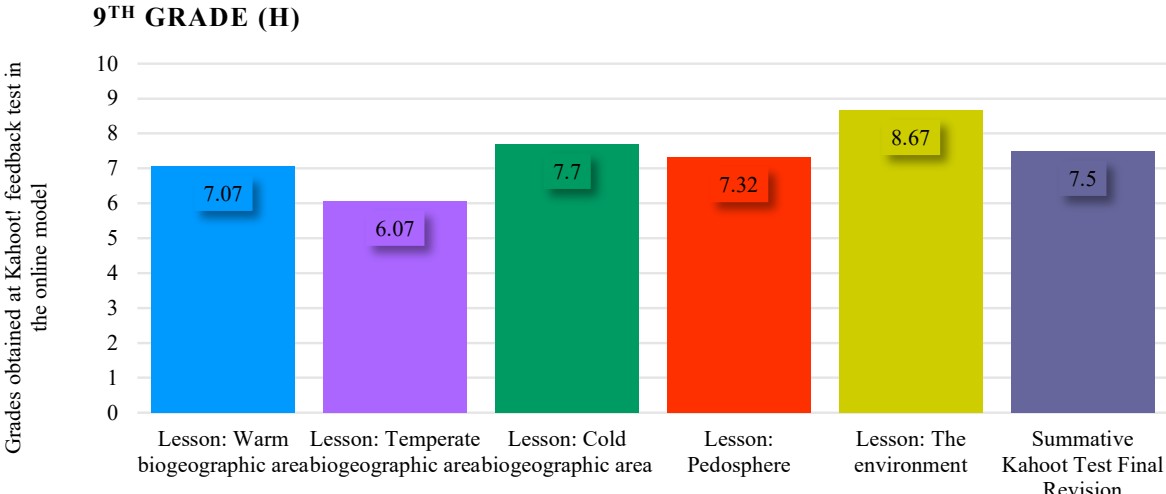

**Figure 7.** Evolution of student grades in the assessment sequence of the Kahoot! interactive game; type—traditional model of teaching–learning–assessment—9th Grade parallel classrooms A, C, D, E, F, G, H.

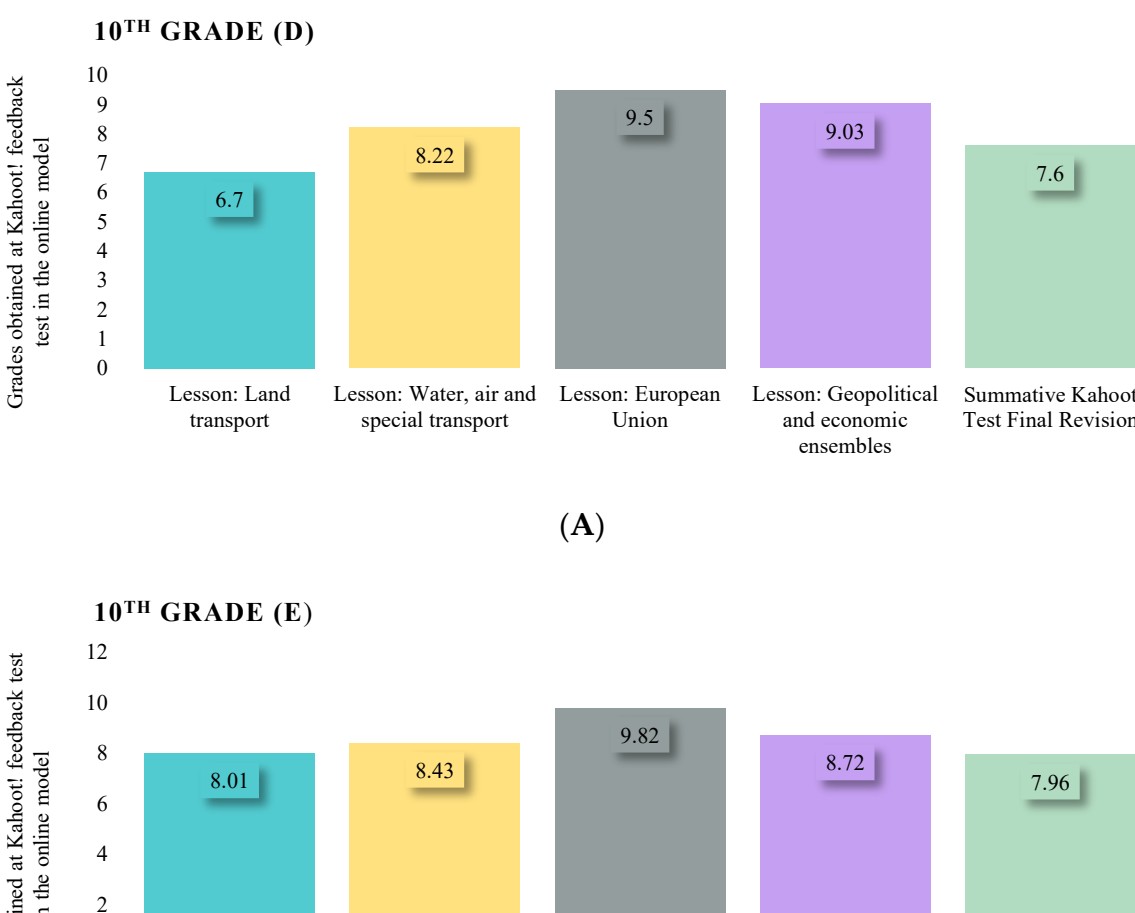

**Figure 8.** *Cont.*

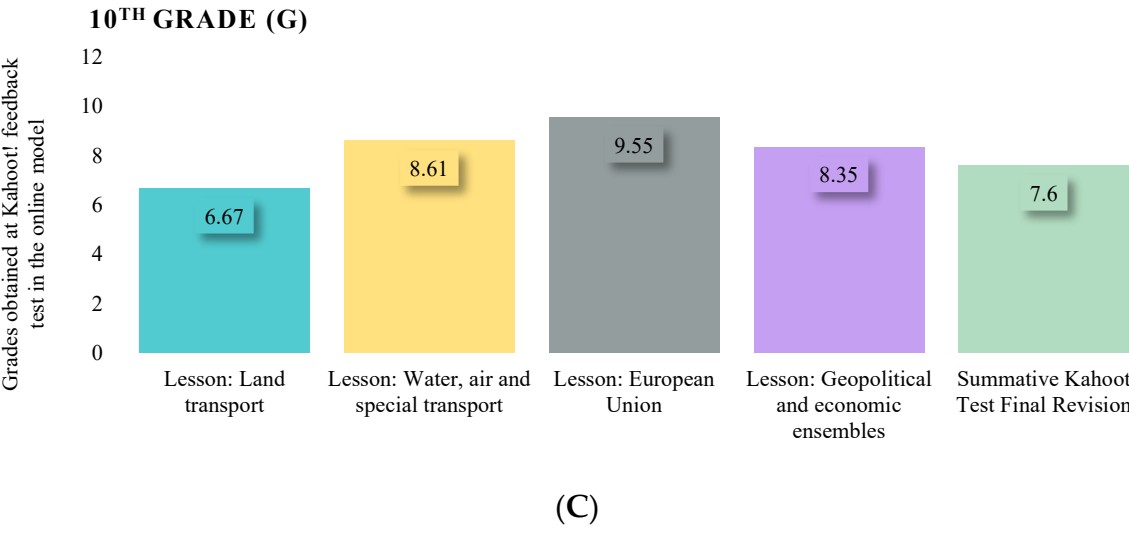

**Figure 8.** Evolution of student grades in the assessment sequence of the Kahoot! interactive game; type—traditional model of teaching–learning–assessment—10th Grade parallel classrooms D (**A**), E (**B**) and G (**C**).

During the research with the two comparative teaching and learning models, in terms of the values of the impact size per class for the students′ school results based on a standardized assessment background, the comparative values between the classes were as follows: in the fifth grade, the averages per class on the online model (Figure 1) were 7.64 (5th grade A) and 6.29 (5th grade B) compared to the traditional model (Figure 5), which had averages of 6.96 (5th grade A) and 7.14 (5th grade B). It must be noted that in the Romanian education system, multiple classes of the same grade are assigned letters—thus 5th grade A and 5th grade B are both the same grade, but the students learn separately.

Additionally, in the 6th grade, the averages per class on the online model (Figure 2) were 6.79 (6th grade -A) and 6.53 (6th grade B) compared to the traditional model (Figure 6), which had averages of 7.88 (6th grade -A) and 6.86 (6th grade B). There is a significant increase in the grades of the students learning in the traditional model compared to the online one, with a maximum value of 1. 09 points in 6th grade A (13.9%), 0. 33 points in 6th grade B (4.9%), and 0.85 points in 5 grade B (12%).

In the 9th grade, the averages per class in the online model (Figure 5) were 6.88 (9th grade A), 6.32 (9th grade C), 8.34 (9th grade D), 7.00 (9th grade E), 8.20 (9th grade F), 7.13 (9th grade G), 8.04 (9th grade E), 8.04 (9th grade E), 8.04 (9th grade E), 8.20 (9th grade F), 7.13 (9th grade G), 8.04 (9th grade G), and 8.04 (9th grade H) compared to the traditional model (Figure 6), which had averages of 7.49 (9th grade A), 7.69 (9th grade C), 7.97 (9th grade D), 7.80 (9th grade E), 8.07 (9th grade F), 8.07 (9th grade G), and 7.36 (9th grade H).

In the 10th grade, it can be noted that the averages per class studying in the online model (Figure 4) were 6.89 (10th grade D), 7.24 (10th grade E), and 7.51 (10th grade G), compared to the traditional model (Figure 8), which had averages of 8.21 (10th grade D), 8.58 (10th grade E), and 8.15 (10th grade G).

In high school classes, a spectacular increase in the results can be observed in the traditional model compared to in the online model in the 9th and 10th grades, with a maximum value of 1.37 points in 9th grade C (17.9%) and of 1.32 points in 10th grade D (16.1%).

The traditional model has been observed to be better than the online model in the context of grades. The positive influence regarding the traditional model is the fact that all students are carefully supervised by the teacher, all of the students take notes under direct observation and guidance, and all of the students carry out the proposed learning activities, thus improving the students′ results and highlighting their progress. This results in a higher percentage of active students compared to passive students when compared to the online model.

In the online teaching–learning system, a negative factor is represented by the poor direct supervision of the students by the teachers, the students being able to have completely different concerns during the teaching–learning management sequence, resulting in random and wrong answers during the assessment period, thus generating the lower results.

The Google Forms, active method, was combined with PowerPoint presentation, a passive method, to create a method that could be used to verify the knowledge that the students retained from the presentation, and this combined method was more effective than in cases where only the passive PowerPoint method was used. Additionally, during the face-to-face period, the summative test, where all of the content studied throughout the year is applied, the students obtained slightly lower grades. The reason for this is that students cheat less in the traditional model; thus, the grades that they obtain are closer to their current level of knowledge.

Other causes of lower results using the Kahoot method! can be caused by the poor internet connection that some students have, the degree of difficulty of the contents of the lessons being studied, and the hardware performance of the devices used.

These results highlight the fact that the variable influencing the results is the training approach, which must be rethought so that all of the students are involved in the educational process since the students who were inattentive during the teaching process or who did not formulate answers to the learning activities during the class but who participated in the Kahoot! game assessment test achieved very poor results.

The results obtained by each class in the third stage of the Google Forms questionnaire on the student perceptions of the interactive game of Kahoot! for the feedback-back/summative assessment sequence in geography lessons are shown in Figure 9.

Thus, the minimum and maximum average values are 100% for the total disagreement scale and for student disagreement with the three questions in Figure 9; 84.9% total agreement for the question "To what extent do you consider that the interactive Kahoot! game for feed-back/summative assessment has a positive effect on learning geography?; 77% agreement for the question "To what extent do you consider that the Kahoot! interactive game of feed-back/summative assessment is a motivating tool for teaching activities in geography classes?"; and 87.3% agreement for the question "To what extent do you think you want to use the interactive game Kahoot! in geography lessons as a feedback/summative assessment sequence in the next school year?"

**To what extent do you think that the Kahoot! interactive feedback / summative assessment game has a positive effect on geography learning?**

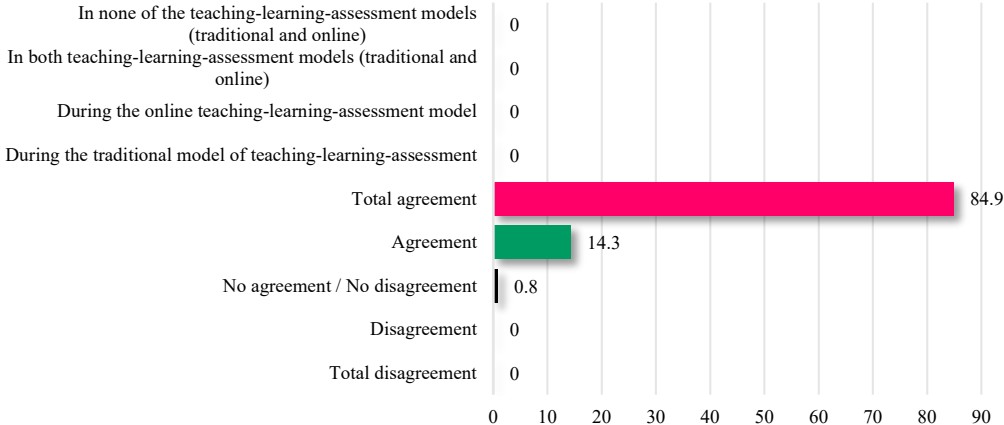

**Figure 9.** *Cont.*

**To what extent do you consider the Kahoot! interactive feedback / summative assessment game to be a motivating tool for teaching geography?**

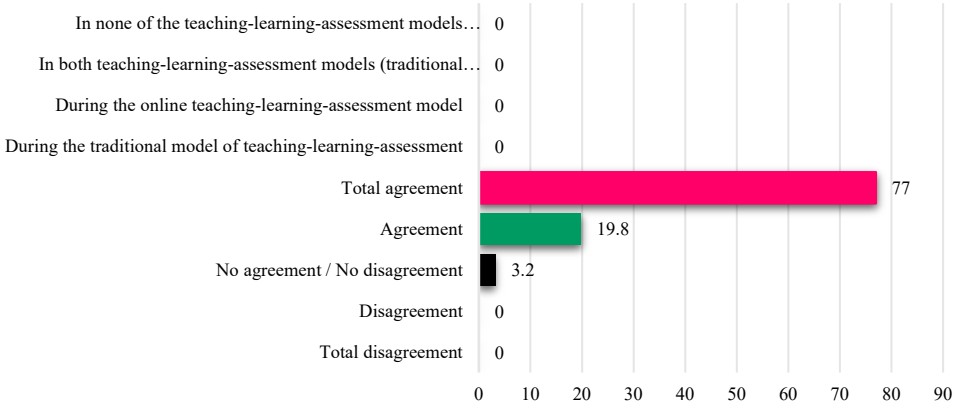

**To what extent do you think that next school year you want to use the interactive game Kahoot! in the geography lessons as a sequence of feedback / summative evaluation?**

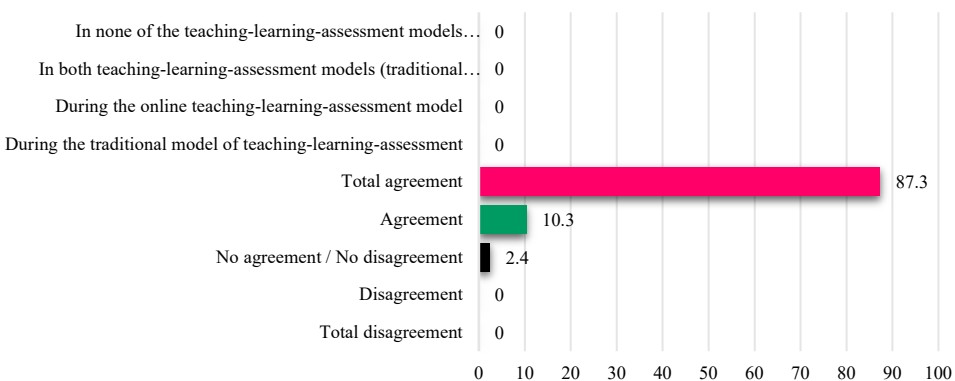

**Were you more attentive and better at the interactive Kahoot! feedback / summative assessment game in the online model or in the traditional teaching-learning-assessment model?**

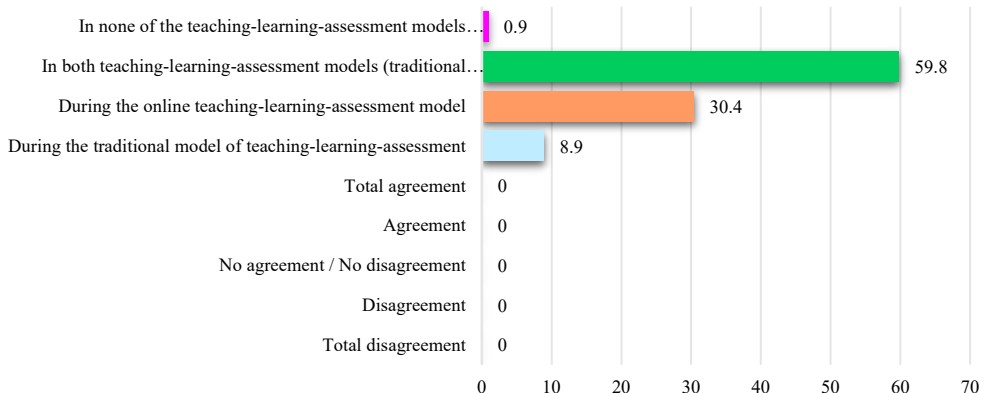

**Figure 9.** Student perception of the interactive game Kahoot! for the feedback/summative assessment sequence in geography lessons.

Regarding the question "Have you been more attentive and achieved better results during the Kahoot! interactive feed-back/summative assessment game in the online model or in the traditional teaching–learning–assessment model?", there was a minimum value of 0.9% in the response variant in both of the two teaching–learning–assessment models, with a maximum value of 59.8% attention and better grades being achieved for both teaching–learning–assessment models.

In terms of whether students desired to use the interactive game Kahoot! in future geography lessons and why, multiple reasons stood out: the Kahoot! platform is a motivating tool for study; develops competitiveness between students in the class and other classes; increases grades and attention; provides more response options; is an interesting and fun, way to learn and consolidate the information of the lesson in class; provides feedback to the teacher about who understood the lesson and who did not; and it is easier to learn through play. The students also stated that Kahoot! helped the students pay attention; the scores were a motivational factor; the game is simple to use/accessible; it has a reduced assessment time; it is a method of real-time self- assessment; it develops thinking skills; it is relaxing; it helps to conduct an interactive lesson; it is a practical, interactive, and innovative revision method and effective learning method; it is creative; it offers distributive attention and answers for all students; and it is easier than conducting a test on paper. Additionally, the students that the Kahoot! game offers a friendly type of learning; has a three-person podium that is motivating because of the social desire to be at the top; provides a ranking for all students; is educational; stimulates students to complete a productive activity; is interesting; provides helpful images; has a positive effect on the performance of students; does not promote cheating; and improves digital skills. The objectivity of the assessment of the results and the development of self-confidence induces a state of well-being.

## 4. Discussion

The main purpose of the research was to check to what extent the interactive game Kahoot! helps students to progress and be motivated and interested in the study of geography. It has also been shown that in the traditional model, the supervision and involvement of all students in the teaching–learning process leads to higher performance, and in the online model, the Google Forms tool method combined with a PowerPoint presentation led to the students paying attention and being motivation to learn and develop the competences in the curriculum. Formative assessment is what supports and promotes learning [38,39].

The research results showed that the method of assessment based on the use of the Kahoot! platform improves the quality of the teaching–learning–assessment process and encourages all students to participate and take on an active role in class. Therefore, the choice of the teacher regarding the teaching–learning–assessment method has a direct impact on the knowledge retained by the students. [40,41].

This study also highlighted that the standardized assessment of student outcomes using the interactive exercise Kahoot! has positive effects on learning, which is in agreement with, other studies [34] as well as on student perceptions [35,36].

This study can contribute to the completion of practical approaches regarding effective methods of assessment as well as practice approaches for teaching and learning in the pre-university environment.

The current study has limitations that need to be considered, but that could indicate potential future lines of research. One of the main limitations is the size and origin of the sample (392 students of a single Romanian high school); therefore, generalizations based on these results should be treated with caution.

## 5. Conclusions

This study aimed to prepare a list of interventions to improve types of formative assessment with the Kahoot! interactive exercise followed by the analysis of a questionnaire

determining student perceptions regarding the reuse of the Kahoot! platform as the form of assessment and feedback in the final revision sequence in the next school year.

The Kahoot! game learning platform used in the assessment sequence induces neither a separation from the operational objectives of the lesson and the specific learning competencies defined in the curriculum and does not require or imply special digital skills.

The efficiency of the formative/summative assessment modality with the Kahoot! interactive game requires patience, constant and increased attention, a permanent and qualitative internet connection as well as two electronic devices (preferably for those who only benefit from the phone).

It is highlighted that the motivation, ease, and usefulness perceived by students regarding the use of the interactive game Kahoot! have a positive and significant influence on the training of the skills indicated in the school curriculum, creating an environment that is conducive to study. This method can be used in all three teaching–learning models (traditional, hybrid, and online), thus ensuring that the method remains consistent in any scenario, reducing student stress during assessments.

Recent studies on the impact of COVID-19 on educational institutions are relatively few; however, more research has proposed ways in which the transition from face-to-face teaching to online learning could be achieved, for example, digital literacy training, the use of online classrooms, encouraging students to use peer learning, and teachers focusing on improving the virtual or physical engagement of their students [42–45].

The results of this study will be of interest to the specialized field literature, providing an overview of the positive results in terms of the learning and perceptions of students on the value of the Kahoot! platform as a method of evaluation in pre-university education.

The fundamental changes that have occurred in Romanian education system requires concrete actions in order to increase the adaptive capacity of the educational system. One of the directions of action to overcome this challenge can be represented by the students' knowledge of new methodologies for the analysis of geography. Knowledge of the basic elements of the Geographic Information System (GIS) and fractal analysis can be easy means of transmitting new trends in modeling the territorial reality [46–53].

Possible future research directions could include the application of the proposed learning platform in other cultural, geographical, and economic areas and in the context of systemic crises.

**Author Contributions:** Conceptualization, F.T. and D.C.D.; Methodology, F.T. and D.C.D.; Validation, D.C.D.; Formal analysis, F.T.; Investigation, F.T.; Data Curation, F.T.; Writing—original draft, F.T., D.C.D. and C.M.P.; Writing—review & editing, F.T., D.C.D. and C.M.P.; Visualization, D.C.D.; Supervision, D.C.D. All authors have read and agreed to the published version of the manuscript.

**Funding:** This research received no external funding.

**Institutional Review Board Statement:** Not applicable.

**Informed Consent Statement:** Informed consent was obtained from all subjects involved in the study.

**Data Availability Statement:** All supporting data is available in the article.

**Conflicts of Interest:** The authors declare no conflict of interest.

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
