# Peer review of "The Use of the Kahoot! Learning Platform as a Type of Formative Assessment in the Context of Pre-University Education during the COVID-19 Pandemic Period"

_education, doi:10.3390/educsci11100649_

Round 1

Reviewer 1 Report

This paper presents a series of experiments on the application of kahoot as a platform for teaching a secondary school geography course in Romania.

As mentioned in the paper, the kahoot platform has been in development for almost 10 years, which means that it has actually been developed by academics on a certain scale.

However, this article has good implications for teaching and learning discussions in Romania, while my suggestions for this article are:

1. The first part of the article is too concise and does not even introduce the relevant background of Romania, which needs to be added.
2. The middle part of the article is a huge stack of data, which could instead be considered for streamlining

Author Response

Dear reviewer

Thank you for your comments. 

All the best,

Reviewer 2 Report

The author(s) presume that everyone knows what the Kahoot platform is. However, the ms would be improved by telling readers exactly what it is and why it is important.

The language need to be more formal,. See, e.g., line 60 "a lot" is  too informal, almost slang....better to say a great deal....

The discussion should be expanded to discuss practical consequences.  

Further, can the author(s) offer ant practical suggestions of what this means in practice?

Author Response

(The authors gave the same response as above.)

Reviewer 3 Report

General comment: Paper is presented as Article but focuses on the use of Kahoot! and results in grades of Geography lessons.

Introduction reduces to assert what some papers/researchers obtain but there is no discussion or clear line of reasoning and making clear if the authors agree with some or none of them.

Sections 2 contains the Methods and 3 the Results? along 12 pages with 9 figures included. Sections 4 and 5 retake two words/issues from the title (formative and COVID-19) a couple of times as highlighted below.

Some concrete suggestions:

The authors should revise the English and make sure that Kahoot!, feedback and Figure n is properly written all around.

There are several vague statements (and sometimes categorically stated!) without any reference or citation to support them, this should be fixed. Some of them have been underscored in the pdf.

The authors do not explain the A,B,C,D,.. of the different Grades, they should be consistently written in format and avoid repetitions. Anywhere these results are merely descriptive of the grades obtained and there is no explanation or clear conclusion.

Title contains reference to COVID-19 pandemic. It is cited once in Abstract, 3 times in Introduction (poorly referenced in L44-46 and missing a lot of going on research on the topic). The same applies to its final reference in L 275-277.  A lot has been written on COVID-19 adaptation at all levels of teaching and learning.  The authors do not consider the case in which blended learning was on stage before the pandemic which seem to have smoothened the upcoming of the forced online teaching because of the pandemic as exhibited in

Sánchez-Ruiz, L.M.; Moll-López, S.; Moraño-Fernández, J.A.; Llobregat-Gómez, N. B-Learning and Technology: Enablers for University Education Resilience. An Experience Case under COVID-19 in Spain. Sustainability 2021, 13, 3532. https://doi.org/10.3390/su13063532

Title contains the word formative assessment: a similar situation happens to COVID-19. It is cited once in the Abstract, once in the Introduction (L61). And it does not come back till the discussion (L237) and twice in the Conclusion (L259 and 266).

There is no description of how summative / formative is Kahoot!, if all final grading comes from Kahoot! Or, otherwise, what its weight is. This should be clarified. In addition it is clear that this formative/summative assessment is a form of continuous assessment and some mention to it migh be in place, cf. https://doi.org/10.3390/math9172082

The interesting part of the use of Kahoot! should be visible to include the Questionaires that the students have answered.

In the Introduction, the authors should make clear what is the added value of this paper compared to the existing literature, its aim and the reason why it should be of interest to read.

In addition to the general and concrete above comments there are some repetitions, unclear sentences or categorical statements in the text that have been highlighted and the authors should revise, rephrase or provide adequate references.

Author Response

(The authors gave the same response as above.)

Round 2

Reviewer 1 Report

Thank you for the modifications, I don't have any other better comments to offer in this case.

Author Response

Dear reviewer,

I have attached below the pdf file with the response to your suggestions.

Thank you for your assistance in improving the scientific relevance of this article.

All the best,

Reviewer 2 Report

The paper is certainly improved and is acceptable in its present form.

Still, based on the findings, I think it would make the paper stronger if it were to add some recommendations for practice. In other words, tell readers what they should do to learn from the paper.

Author Response

(The authors gave the same response as above.)

Reviewer 3 Report

Paper is clearer and English has improved but still requires reviewing, mainly in all highlighted words/expressions of the uploaded file to improve style, avoid repetitions and facilitate its understanding.

Some native speaker should revise the whole text as some sentences are extremely long or hard to follow.

For instance L34-34 and L40-41 are vague somehow and require clarification if kept.

The authors state that the aim of the paper is to OPTIMIZE... That word  should change over all the text as optimizing has a meaning associated with getting an OPTIMAL result which is not the case... 

Citing has improved but some care must be taken to avoid sentences with "A study...", "An artucle studied...", "A recent study...", "Resurch exists...". In general terms credit by naming the authors of the contributions should be followed, especially when there is only one source and opinion is relevant and by using the "et al." form within text when a paper has more than two authors.  One more cite is suggested in L252.

L286 assertion should be revised as indeed there is a wide literature on COVID-19 from different perspectives and countries. Its transtition in L287 also requires some rephrasing.

Author Response

(The authors gave the same response as above.)

Round 3

Reviewer 3 Report

Output has improved.

The authors answer that they have replaced the word optimise by "improve". They missed L12, 91, 269 where the word optimize/optimal remains.

Abstract might yet delete some sentences and get some rephrasing so that it is more concise and does not lose its informative message. Next is just a suggestion and authors should decide if really the Abstract needs to be longer.

The present study aims to display how using a personal assessment environment based on the Kahoot! interactive exercise is actively supporting the teaching-learning process. The goal is to improve the instructive-educational process by applying a learning platform based on play and digital technology, which favors a qualitative educational endeavor. The use of the Kahoot! platform as an assessment form had a significant direct positive effect on the educational process during the COVID-19 pandemic. 

Author Response

Dear reviewer

Thanks for the help in improving the article.

All the best
